# Assessing private provider perceptions and the acceptability of video observed treatment technology for tuberculosis treatment adherence in three cities across Viet Nam

**Lan Huu Nguyen[1], Phuong Thi Minh Tran[2], Thu Anh Dam[2], Rachel Jeanette Forse[2], Andrew James Codlin[2], Huy Ba Huynh[2], Thuy Thi Thu Dong[2], Giang Hoai Nguyen[3], Vinh Van Truong[1], Ha Thi Minh Dang[1], Tuan Dinh Nguyen[4], Hoa Binh Nguyen[4], Nhung Viet Nguyen[4], Amera Khan [5], Jacob Creswell[5], Luan Nguyen Quang Vo [2,3]***

**1** Pham Ngoc Thach Hospital, Ho Chi Minh City, Viet Nam, **2** Friends for International TB Relief, Hanoi, Viet Nam, **3** IRD VN, Hanoi, Viet Nam, **4** National Lung Hospital, Hanoi, Viet Nam, **5** Stop TB Partnership, Geneva, Switzerland

* luan.vo@tbhelp.org

**Data Availability Statement:** All relevant data are within the paper and its Supporting information files.

## Abstract

### Background

The World Health Organization recently recommended Video Observed Therapy (VOT) as one option for monitoring tuberculosis (TB) treatment adherence. There is evidence that private sector TB treatment has substandard treatment follow-up, which could be improved using VOT. However, acceptability of VOT in the private sector has not yet been evaluated.

### Methods

We conducted a cross-sectional survey employing a theoretical framework for healthcare intervention acceptability to measure private provider perceptions of VOT across seven constructs in three cities of Viet Nam: Ha Noi, Ho Chi Minh City, and Hai Phong. We investigated the differences in private providers' attitudes and perceptions of VOT using mixed ordinal models to test for significant differences in responses between groups of providers stratified by their willingness to use VOT.

### Results

A total of 79 private providers completed the survey. Sixty-two providers (75%) indicated they would use VOT if given the opportunity. Between private providers who would and would not use VOT, there were statistically significant differences (p≤0.001) in the providers' beliefs that VOT would help identify side effects faster and in their confidence to monitor treatment and provide differentiated care with VOT. There were also significant differences in providers' beliefs that VOT would save them time and money, address problems faced by their patients, benefit their practice and patients, and be relevant for all their patients.

**Funding:** This research was supported by the TB REACH initiative of the Stop TB Partnership, grant number STBP/TBREACH/GSA/W5-25, with funding from the Global Affairs Canada.

**Competing interests:** The authors have declared that no competing interests exist.

**Abbreviations:** DOT, Directly Observed Treatment; FIT, Friends for International Tuberculosis Relief; HCMC, Ho Chi Minh City; NTP, National TB Control Program; PPM, Public-Private Mix; PPM-DOTS, Public-Private Mix; Directly Observed Treatment, Short-course; TB, Tuberculosis; VOT, Video Observed Treatment; WHO, World Health Organization.

## Conclusion

Private providers who completed the survey have positive views towards using VOT and specific subpopulations acknowledge the value of integrating VOT into their practice. Future VOT implementation in the private sector should focus on emphasizing the benefits and relevance of VOT during recruitment and provide programmatic support for implementing differentiated care with the technology.

## Introduction

In Viet Nam, an estimated 174,000 people developed tuberculosis (TB) in 2019 and there were 13,200 deaths from the disease, despite free treatment being readily available at government health facilities [1]. The country's National TB Control Program (NTP) has successfully increased the number of TB cases notified, linked more individuals to treatment, and achieved high treatment success rates. The NTP attributes these achievements to robust nationwide coverage of treatment services that included quality-assured diagnostics and standardized treatment regimen with supervision and patient support as well as facility-based directly observed treatment (DOT) as per national treatment guidelines [2].

While studies have found DOT can improve treatment outcomes [3], a more nuanced understanding of treatment observation has emerged in recent years. There are often conflicting views on whether DOT actually results in better treatment outcomes as compared to unsupervised, self-administered treatment. In an attempt to reproduce previous findings on the impact of DOT, McKay et al. found a small, non-statistically significant difference between DOT and self-administered treatment, along with a lack of high-quality evidence to compare these TB treatment methods in the form of prospective cohort or randomized trials [4]. A recent patient costing survey in Viet Nam showed that 63% of people being treated for drug-susceptible TB experienced catastrophic costs (>20% of annual household income spent on the patient's episode of TB), and the majority of out-of-pocket spending occurred during TB treatment [5]. In recent years, WHO has endorsed a more "patient-centered approach to TB care" and recommended that TB interventions explore other treatment administration options, such as digital technologies to provide more individualized treatment support [6,7].

The implementation of DOT in high-TB burden, low-resource settings also often faces limitations; facility-based DOT requires patients to regularly travel to health facilities, resulting in high costs [8] and the loss of autonomy, privacy and time [9]. Consequently, almost all private-sector treatment sites offer patients self-administration of their medication as an incremental value proposition over NTP's services [10]. This presents a particular challenge for TB management, as irregular TB treatment adherence—even among patients who achieve treatment completion—is associated with disease relapse and the development of drug resistance [11,12].

In an attempt to address these barriers, recent innovations in technology have sought to make treatment and adherence support more people-centered. Video observed treatment (VOT) is a derivation of DOT in which the observation of dose taking is achieved with either real-time video (synchronous) or recorded (asynchronous) video upload [13]. The asynchronous method of treatment management allows patients to record and transmit a video of themselves ingesting their medication to a central platform. These videos are later reviewed by healthcare workers. This technology eliminates the need for healthcare workers and patients to be in the same physical space and allows for virtual observation at times that are convenient

for both the patient and the treatment supporter. These digital approaches have been shown to reduce costs and barriers associated with treatment adherence for patients and has resulted in a higher percentage of completed treatment doses [14,15]. Recent studies comparing DOT to VOT have found the latter to be more effective (achieves higher treatment completion rates) and acceptable (achieves higher patient satisfaction) as a treatment observation tool for both active and latent TB [16–18].

VOT implementation in various countries has been shown to be acceptable and feasible in programmatic settings [19,20] and patients enrolled in VOT have similar treatment outcomes compared to those enrolled on DOT [14,18,21]. A number of studies have also found VOT to be effective among patients with drug-resistant TB strains who typically suffer poorer treatment outcomes and require closer monitoring [22–24]. One study on VOT was conducted in Viet Nam with patients from NTP clinics in Hanoi. It utilized an asynchronous platform (SureAdhere) and showed that VOT was feasible and resulted in high treatment adherence in a resource-limited setting [21]. Based on the evidence from these studies, the WHO recommended VOT as a viable alternative to DOT [25]. In light of these recommendations and the successful use of VOT in groups with higher risk of being lost to follow-up, such as drug-resistant TB patients, there is an opportunity to expand its use to other areas with consistently poor treatment outcomes.

One of these areas is the private sector. Studies in Viet Nam have found that less than half of TB patients complete treatment in the private sector, compared to about 90% in the public sector [1,26]. Commonly cited reasons are lack of follow-up and case holding [26–28]. As a consequence, the private sector is thought to be one of the main sources of the development of drug-resistant forms of TB [29]. This emphasizes the necessity of improving patient care and treatment adherence within the private sector.

Past studies in Viet Nam suggested that more than half of the people with TB 'missed' by the NTP in urban areas are seeking care in the private sector [30]. The private sector represents the first point of contact for 70% of persons with TB, and 40% of TB patients are treated exclusively by the private sector [31]. A private provider interface agency model piloted in Ho Chi Minh City (HCMC) between October 2017 and March 2019 discovered 1,112 TB patients being treated in the private sector in two districts of the city [32]. As such, alongside case management, VOT may have the secondary benefit of elucidating the TB burden managed by the private sector through documented treatment monitoring.

VOT can offer tangible public health benefits to TB programs if successfully employed in the private sector. There are many studies that explore patient experiences and attitudes towards mobile health innovations; however, a patient's initial exposure and attitude towards a new program or intervention is greatly influenced by the way a provider presents the opportunity [33,34]. However, our literature review produced no published studies on provider perceptions of VOT for TB patients, particularly for those in the private sector. As such, this study aimed to conduct formative research to assess the acceptability of VOT among private providers for improved case management and follow-up of their clientele.

## Methods

### Study design

This was a cross-sectional study implemented from January to September 2019 in three cities across Viet Nam: Ha Noi, Ho Chi Minh City (HCMC), and Hai Phong.

### Participant eligibility and recruitment

The target population consisted of private providers actively engaged in TB care activities. Providers were eligible to participate if they worked in private practice, treated TB patients, and

provided informed consent. The calculated sample size was 70 based on a 2-sample, 2-sided comparison of mean scores on a 5-point Likert scale with a 95% confidence level and 80% power. We estimated an average rating difference between private providers of 0.9 and a population variance of 4 based on initial results from piloting the survey instrument and intentionally oversampled for the eventuality of post-hoc exclusion of unclear or invalid responses. We invited all eligible private providers who were partnering with Friends for International Tuberculosis Relief (FIT) on the scale-up of the private provider interface agency model [32]. We additionally used chain-sampling; interviewees and other health care workers who were recruited for our study referred us to other private providers who might be eligible to participate. We approached clinicians who either themselves confirmed, or whom other doctors indicated, provided private TB treatment in their clinics. Directors and owners of private clinics with decision-making power regarding the use of VOT in their clinics were also included in the study.

## Data collection and processing

We developed a VOT acceptability survey consisting of 29 questions in English (Survey in S1 Survey) and translated it into Vietnamese (Survey in S2 Survey). The development of the survey is based on Sekhon et al's theoretical framework of acceptability (TFA) [35]. This framework measures acceptability of healthcare interventions along eight constructs including: 1) ethicality; 2) intervention coherence; 3) burden; 4) opportunity cost; 5) perceived effectiveness; 6) self-efficacy; 7) usability/ implementation; and 8) affective attitude [36]. We chose to develop our own survey due to the absence of any internationally validated instrument on the acceptability of mobile health technologies for TB. The survey questions utilized a 5-point Likert scale ranging from strongly disagree to strongly agree or from very difficult to very easy.

To test the reliability of the survey instrument, we calculated Cronbach's alpha for internal consistency. The coefficient rate of more than 0.70 is generally considered to be acceptable. We found a high degree of internal consistency for each survey question ($\alpha \geq 0.859$; detailed results included in the Table in S1 Table).

Besides assessing each acceptability construct, the survey bifurcated respondents through a dichotomous question around willingness to try VOT and collected demographic information on age, gender, years since the completion of medical training, length of time working in a private clinic and whether the provider also worked in the public sector.

A medical doctor and study staff familiar with treating TB in the private sector both verified the questions and piloted the survey. All staff conducting the interviews were trained on survey methods and good clinical practices before interviewing participants.

Prior to fielding the survey, participants were introduced to the SureAdhere VOT platform (San Diego, USA), an asynchronous smartphone application, and were shown a 2-minute video demonstrating how VOT is used by patients (https://youtu.be/DiE-3a1GTn0). The survey was fielded using the open-source ONA platform on Android tablets (https://ona.io/).

After survey responses were collected in ONA, the data were cleaned and checked for completeness at the end of each week. We conducted additional follow-up interviews in the event of missing data. When a participant did not know or refused to answer a question, the response was excluded from the analysis.

## Statistical analyses

We calculated means, medians, or proportions with corresponding 95% confidence intervals scores for each response. We then used the Chi-squared and Fisher's exact tests to identify significant differences in the demographic factors and responses to acceptability constructs

between the three cities. If these tests were statistically significant, the Holm-Bonferroni sequential correction was applied to account for multiple comparisons. The medians and means of the Likert-scale responses were stratified by the providers' willingness to use VOT. We fitted mixed ordinal proportional odds models [37] to explore the association between individual acceptability constructs and providers' willingness to use VOT as the fixed effect variables, and city as the random effect. In the event of small or empty cells in individual groups, we aggregated individual 5-point Likert scale responses into three categories to stabilize the model: 1) Disagree, 2) Neither disagree nor agree, and 3) Agree. When the assumption of proportional odds was violated based on comparing models both assuming and relaxing the proportional odds assumption, we used mixed non-proportional odds models. When we compared the willingness to use VOT among private providers across the three cities, a stratification by city was not possible due to homogeneity in responses from participants in Hai Phong. Therefore, we conducted additional Mann-Whitney *U* tests using the varied acceptance data in HCMC only (Table in S3 Table), as well as for the combined data from Ha Noi and Hai Phong (Table in S5 Table). Statistical analyses were performed using Stata v14 (College Station, TX: StataCorp).

## Ethical considerations

Ethical approval for this study was granted by the Ethics Committee for Biomedical Research of the Ha Noi School of Public Health (324/2019/YTCC-HD3). All participating providers were given a participant information sheet and signed an informed consent form prior to completing the survey. We removed all personally identifiable information from the survey responses prior to the analyses.

## Results

We invited 101 eligible private providers to participate, which included 85 single-doctor clinics and eight private hospitals or multi-clinics. Overall, 79 (78%) private providers agreed to participate in the study, with 65 of these providers working in both the private and public sectors. Of the 79 study participants, 28 out of 31 eligible private doctors (90%) were located in Hai Phong, 17 of 25 eligible private doctors (68%) were located in Ha Noi, and 34 of 50 eligible private doctors (68%) were located in HCMC. Across the three cities, 17 (22%) participants had pre-existing relationships with FIT: 8 (24%) in HCMC, 2 (11%) in Ha Noi and 7 (25%) in Hai Phong. Reasons for declining to participate included worries about confidentiality, having insufficient time for an interview, and being uninterested in VOT implementation.

## Demographics

Table 1 below provides demographic information on private providers who completed the survey. 63% (n = 50) of private providers surveyed were male, 66% (n = 52) of private providers were 40 years or older, and 60% (n = 47) graduated from medical school more than 15 years ago. Of participants surveyed, 59% (n = 47) had practiced in the private sector for more than 5 years, and 82% (n = 65) of providers worked in both the private and public sectors.

There was a significant difference in the distribution of gender and age of private providers between the three cities. While 82% (n = 28) of providers surveyed in HCMC were male, only 41% (n = 7) and 54% (n = 15) of respondents were male in Ha Noi and Hai Phong, respectively. Over two-thirds of private providers in HCMC were aged 40 years or older; the proportion was about 50% in Ha Noi and Hai Phong.

**Table 1. Characteristics of private providers in the survey stratified by city.**

|  | Total (N = 79) (N, %) | HCMC (N = 34) N, % | Hai Phong (N = 28) (N, %) | Ha Noi (N = 17) (N, %) | P-value |
|---|---|---|---|---|---|
| **Sex** |  |  |  |  |  |
| Female | 29 (37) | 6 (18) | 13 (46) | 10 (59) | **0.007**[$] |
| Male | 50 (63) | 28 (82) | 15 (54) | 7 (41) |  |
| **Age** |  |  |  |  |  |
| 28–40 years | 27 (34) | 5 (15) | 15 (54) | 7 (41) | **0.021**[$] |
| 41–50 years | 21 (27) | 13 (38) | 5 (18) | 3 (18) |  |
| 51–71 years | 31 (39) | 16 (47) | 8 (29) | 7 (41) |  |
| **Years since medical school**[*] |  |  |  |  |  |
| 3–15 years | 32 (41) | 11 (32) | 15 (54) | 6 (35) | 0.174 |
| 16–30 years | 29 (37) | 17 (50) | 7 (25) | 5 (29) |  |
| 31–41 years | 18 (23) | 6 (18) | 6 (21) | 6 (35) |  |
| **Years with private clinics** |  |  |  |  |  |
| 1–5 years | 32 (41) | 12 (35) | 14 (50) | 6 (35) | 0.444 |
| 6–31 years | 47 (59) | 22 (65) | 14 (50) | 11 (65) |  |
| **Employment sector** |  |  |  |  |  |
| Mix of public and private sectors | 65 (82) | 29 (85) | 22 (79) | 14 (82) | 0.788 |
| Private sector only | 14 (18) | 5 (15) | 6 (21) | 3 (18) |  |

[*]Percentages may not total 100 due to rounding.

[$]Statistically significant difference after adjusting for multiple comparisons using the Holm-Bonferroni sequential correction.

## Acceptability

Table 2 presents the results on whether private providers agreed with survey statements according to the acceptability constructs. A high proportion of total private providers in all cities expressed that they were willing to test new approaches (86%) and agreed or strongly agreed that observation is the best strategy for adherence (86%). Many providers agreed that VOT could identify individuals at risk for stopping treatment faster (82%) and help patients adhere to treatment (82%). Only 16% (n = 13) of private providers thought that VOT is relevant to all patients in their practice. They believed VOT was relevant only to specific populations such as young or educated patients (n = 14) or individuals who often use a smartphone in their daily lives (n = 11).

Across 79 participants, 75% (n = 62) indicated their willingness to use VOT if given the opportunity. Responses to questions under all acceptability constructs except for perceived effectiveness were significantly different across private providers in different cities. Detailed results using the Holm-Bonferroni sequential correction are provided in the supplemental information (S6 Table). While 100% (n = 28) of providers in Hai Phong were willing to use VOT, 88% (n = 18) of providers in Ha Noi and 56% (n = 19) of providers in HCMC were willing. Across all constructs, private providers in Hai Phong had more positive attitudes towards VOT use than in the other cities. About half of the private providers in HCMC agreed that using VOT would save them time (opportunity cost; 53%), address problems which patients face (50%), and be beneficial for their practice and patients (affective attitude; 50%). The percentage of Ha Noi participants who responded 'Agree' to all questions under all constructs was consistently higher than the percentage of participants in HCMC, but lower than the proportion measured among Hai Phong participants. While patient confidentiality was a concern for many providers in HCMC (71%), a lower proportion of providers in Hai Phong (32%) and Ha Noi (41%) expressed these concerns.

**Table 2. Overall attitudes of private providers towards VOT.**

| Constructs of acceptability | Total Agree (N, %) | HCMC Agree (N, %) | Hai Phong Agree (N, %) | Ha Noi Agree (N, %) | P-value |
|---|---|---|---|---|---|
| **Ethicality[a]** | | | | | |
| Belief that observation is the best strategy for adherence | 68 (86) | 24 (71) | 28 (100) | 16 (94) | **0.002**[*][$] |
| Willingness to test new approaches | 68 (86) | 26 (76) | 28 (100) | 14 (82) | **0.044**[*][$] |
| **Intervention Coherence[a]** | | | | | |
| Identify side effects faster | 50 (63) | 18 (53) | 23 (82) | 9 (53) | 0.073 |
| Identify people at risk of stopping treatment faster | 65 (82) | 23 (68) | 27 (96) | 15 (88) | **0.028**[*][$] |
| **Burden (Easy to use)[b]** | | | | | |
| Time requirement from doctor | 33 (42) | 7 (21) | 18 (64) | 8 (47) | **<0.001**[$] |
| Time requirement from patient | 23 (29) | 6 (18) | 10 (36) | 7 (41) | **0.025**[$] |
| **Opportunity Cost[a]** | | | | | |
| Save time for doctor | 55 (70) | 18 (53) | 26 (93) | 11 (65) | **<0.001**[*][$] |
| Save money for doctor | 28 (35) | 8 (24) | 14 (50) | 6 (35) | **<0.001**[$] |
| **Perceived Effectiveness[a]** | | | | | |
| Providing differentiated care | 59 (75) | 22 (65) | 24 (86) | 13 (76) | 0.252[*] |
| Help patients adhere to treatment | 65 (82) | 26 (76) | 24 (86) | 15 (88) | 0.808[*] |
| **Self-Efficacy[a]** | | | | | |
| Confidence in ability to monitor treatment through VOT | 48 (61) | 15 (44) | 22 (79) | 11 (65) | **0.001**[*][$] |
| Confidence in ability to provide differentiated care through VOT | 62 (78) | 26 (76) | 24 (86) | 12 (71) | 0.358[*] |
| **Affective Attitude[a]** | | | | | |
| Addresses problems which patients face | 53 (68) | 17 (50) | 24 (89) | 12 (71) | **0.007**[*][$] |
| Beneficial for doctor's practice and patients | 57 (72) | 17 (50) | 26 (93) | 14 (82) | **0.001**[*][$] |
| Relevant for all of doctor's TB patients | 13 (16) | 3 (9) | 9 (32) | 1 (6) | **<0.001**[*][$] |
| **Implementation/Usability[a]** | | | | | |
| Concerns about patient confidentiality | 40 (51) | 24 (71) | 9 (32) | 7 (41) | **0.011**[*][$] |
| Comfort with receiving support from study staff | 65 (82) | 25 (74) | 26 (93) | 14 (82) | 0.051[*] |
| **Provider willingness to use VOT[c]** | | | | | |
| Yes | 62 (78) | 19 (56) | 28 (100) | 15 (88) | **<0.001**[*][$] |

[a]: 1 (Strongly disagree) to 5 (Strongly agree).

[b]: 1 (Very difficult) to 5 (Very easy).

[c]: Yes/No.

[*]: Fisher's exact test.

[$]: Statistically significant difference after adjusting for multiple comparisons using Holm-Bonferroni sequential correction.

## VOT acceptance and private provider attitudes

Table 3 below shows that between private providers who would and would not use VOT, there are statistically significant differences in the majority of acceptability constructs. The three notable factors on which no significant difference was measured were the perceptions that VOT would help identify people at risk of stopping treatment faster (p = 0.555), time requirement from doctors (p = 0.413) and the doctors' concerns about patient confidentiality (p = 0.544). We have provided stratified analyses for HCMC (Table in S3 Table) and Ha Noi & Hai Phong (Table in S5 Table) in the Supporting information.

## Discussion

Our study provides insight on private providers' perceptions and acceptance of using VOT for providing TB treatment support and to monitor adherence in their practices. The survey tool

**Table 3. Constructs of acceptability associated with the willingness to use VOT in all three cities.**

| Constructs of healthcare intervention acceptability and their components | Would use VOT (N = 62) | | Would not use VOT (N = 17) | | OR (95%CI) | p-value |
|---|---|---|---|---|---|---|
| | Median (IQR) | Mean (95%CI) | Median (IQR) | Mean (95%CI) | | |
| **Ethicality[a]** | | | | | | |
| Belief that observation is the best strategy for adherence | 4 (4–5) | 4.3 (4.0–4.5) | 4 (3–4) | 3.5 (3.0–3.9) | **3.9 (1.0–15.0)** | **0.044** |
| Willingness to test new approaches* | 4 (4–4) | 4.1 (4.0–4.3) | 4 (3–4) | 3.6 (3.2–4.0) | **10.0 (2.5–40.1)** | **0.001** |
| **Intervention Coherence[a]** | | | | | | |
| Identify side effects faster | 4 (3–4) | 3.6 (3.4–3.9) | 2 (2–4) | 2.6 (2.2–3.1) | **7.2 (2.4–21.3)** | **<0.001** |
| Identify people at risk of stopping treatment faster | 4 (4–4) | 3.8 (3.7–4.0) | 4 (3–4) | 3.6 (3.2–3.9) | 1.5 (0.4–5.8) | 0.555 |
| **Burden[b]** | | | | | | |
| Time requirement from doctor | 3 (3–4) | 3.2 (2.9–3.4) | 2 (2–3) | 2.4 (2.0–2.8) | 1.7 (0.5–5.6) | 0.413 |
| Time requirement from patient | 3 (2–4) | 3.0 (2.8–3.2) | 2 (2–3) | 2.3 (1.9–2.8) | **4.6 (1.2–17.0)** | **0.022** |
| **Opportunity Cost[a]** | | | | | | |
| Save time for doctor | 4 (4–4) | 3.7 (3.5–4.0) | 2 (2–4) | 2.8 (2.3–3.2) | **7.0 (1.8–27.4)** | **0.005** |
| Save money for doctor | 3 (2–4) | 3.1 (2.9–3.4) | 2 (2–2) | 2.1 (1.9–2.3) | **27.4 (3.0–249.2)** | **0.003** |
| **Perceived Effectiveness[a]** | | | | | | |
| Help in providing differentiated care | 4 (4–4) | 3.9 (3.7–4.0) | 3 (2–4) | 3.1 (2.7–3.6) | **6.3 (2.1–19.2)** | **0.001** |
| Help patients adhere to treatment | 4 (4–4) | 3.9 (3.7–4.1) | 4 (3–4) | 3.5 (3.0–3.9) | **3.3 (1.0–10.7)** | **0.045** |
| **Self-Efficacy[a]** | | | | | | |
| Confidence monitoring treatment through VOT | 4 (3–4) | 3.7 (3.5–3.8) | 2 (2–3) | 2.6 (2.2–3.1) | **11.7 (3.7–37.2)** | **<0.001** |
| Confidence providing differentiated care through VOT | 4 (4–4) | 3.8 (3.7–4.0) | 3 (2–4) | 3.0 (2.5–3.5) | **10.4 (3.1–35.2)** | **<0.001** |
| **Affective Attitude[a]** | | | | | | |
| Addresses problems which patients face | 4 (4–4) | 3.8 (3.6–4.0) | 2 (2–4) | 2.8 (2.3–3.2) | **8.4 (2.3–31.2)** | **0.001** |
| Be beneficial for doctor's practice and patients | 4 (4–4) | 3.9 (3.7–4.0) | 3 (2–4) | 2.9 (2.4–3.4) | **7.4 (2.0–27.1)** | **0.002** |
| Be relevant for all of doctor's TB patients | 2.5 (2–3) | 2.7 (2.5–2.9) | 2 (2–2) | 2.0 (1.8–2.2) | **6.0 (1.1–33.2)** | **0.042** |
| **Implementation/Usability[a]** | | | | | | |
| Doctor's concerns about patient confidentiality | 3 (2–4) | 3.1 (2.8–3.4) | 4 (2–4) | 3.3 (2.7–3.8) | 1.5 (0.4–5.2) | 0.544 |
| Doctor's comfort with receiving support from study staff | 4 (4–4) | 3.9 (3.8–4.1) | 4 (2–4) | 3.3 (2.9–3.8) | **4.7 (1.5–15.3)** | **0.010** |

[a]: 1 (Strongly disagree) to 5 (Strongly agree).

[b]: 1 (Very difficult) to 5 (Very easy).

*: Individual 5-point Likert scale responses were aggregated into three categories to stabilize the model.

based on the TFA allows for an understanding of acceptability of the intervention prior to its implementation. In the context of three cities in Viet Nam, 78% of providers responded they would be willing to use VOT if given the opportunity. This suggests that the use of VOT in the private sector was acceptable for a subpopulation of providers depending on the value perception for their individual practice.

When bifurcated by providers who would and would not use VOT if given the opportunity across the entire sample, provider responses were consistently positive about the ethicality acceptability construct, highlighting that VOT technology fits well with the providers' value system. Respondents were willing to test new approaches, comfortable receiving support from the program and believed they could implement these technologies successfully given adequate programmatic guidance. Providers saw VOT as effective for aiding patients to adhere to and complete treatment, specifically for patient populations in which mobile phone usage is common.

While all providers in Hai Phong were willing to use VOT, this proportion was 88% in Ha Noi and 56% in HCMC. As such, there seemed to be regional variance in the acceptance of

VOT. Private providers in Hai Phong tended to be younger than in the other cities, particularly HCMC, which may have been a reason for the higher willingness to use VOT [38,39]. Additionally, FIT and other organizations have engaged with many of these private providers in Hai Phong, establishing trust and rapport [40]. This suggests that VOT implementation may be more acceptable in areas where extensive public-private mix (PPM) networks have already been established based on anecdotal evidence as the value proposition of VOT explained by study staff may have been more readily accepted by the private providers.

Meanwhile, provider responses in HCMC showed the highest level of heterogeneity. One factor may have been the share of male participants in HCMC, which was significantly higher than in the other two cities. It has been observed that gender can have a moderating effect on technology acceptance in a developing country setting and that acceptance among men tended to be more associated with perceived usefulness [41]. Meanwhile, women were reportedly more influenced by ease of use and subjective norm [41]. Particularly the latter may have contributed to the higher relative VOT acceptance in Hai Phong. From January–May 2020, we carried out a private provider mapping exercise across Viet Nam and discovered that while there are a substantial number of private providers in the larger cities of Ha Noi and HCMC, the number of private providers in Hai Phong is more limited (200 in Ha Noi, 1,356 in HCMC, and 104 in Hai Phong, unpublished data).

When assessing differences between private providers willing and unwilling to use VOT in HCMC only, the factors that showed a significant difference related to opportunity cost, self-efficacy, and affective attitude. The construct of opportunity cost focuses on the extent to which providers' time and money are affected while engaging in the intervention. Since VOT is an alternative to patients self-administering their treatment, the perception of VOT as efficient and cost-effective within their specific practice was particularly polarizing. Several studies have stated that additional benefits such as laboratory services, free anti-TB drugs, and supporting healthcare workers need to be provided if providers are to adopt a system that requires more effort and is more time-consuming than self-administration [42,43]. Self-efficacy within our study examined providers' confidence in monitoring treatment and providing differentiated care. When providers are comfortable in their ability to use new technology to facilitate achieving their goals, they are more willing to overcome the initial learning curve. The third significant construct of affective attitude encapsulates how an individual feels about an intervention. If providers feel that VOT is beneficial and will help solve a problem in which they face in their clinic, they are more willing to test out the intervention and see more value in the technology.

Providers are fully responsible for managing TB patients in the private sector; thus, the attitudes of private providers play an important role in successfully implementing the intervention. As such, it is important to consider providers' specific obstacles and priorities. This is concordant with studies from other settings, in which private providers were engaged to participate in new approaches in mHealth technology. A study of private providers in India found that worries about confidentiality and lack of computer knowledge demotivated providers from engaging with technological innovations, while proper training on the technology and regular programmatic support acted as enablers [44,45]. Ultimately, the providers' outlook on utilizing mHealth innovations influenced patient adoption of the technology [44].

Our study found that the provider's choice to accept or reject a technological innovation was based on the value proposition of the innovation for their specific practice. Appropriately emphasizing the benefits to their clientele would be critical since several participating private providers raised concern about the relevance of VOT to their patients. Another success factor could be the development of a differentiated care model with clearly defined eligibility criteria under which VOT would be provided to their patients. This is consistent with a qualitative

study involving TB patients which suggests that VOT may not be suitable to all TB patients as the ability to use VOT is limited in some specific groups such as older people who may not be familiar with smartphone technologies [46].

Lastly, acceptability is a dynamic construct subject to current events. Like many other industries, our ongoing private sector engagement has shown that private provider businesses, particularly those treating TB and respiratory illnesses, have suffered heavily from the government's rigorous contact tracing, social distancing and forced quarantine efforts due to the SARS-CoV-2 pandemic. While these efforts have led to Viet Nam's relative success in managing SARS-CoV-2, the deleterious economic impact may have also changed perceptions among private providers to be more accepting of remote care and telemedicine solutions such as VOT. This may particularly be the case in outbreak areas such as Da Nang in July 2020, when patient consultation and follow-up was provided via mobile chat applications during community-wide shut downs.

Studying private provider perceptions has shown us that VOT requires thoughtful planning and application to be successful. The implementation of VOT in a setting where self-administered therapy is the standard of care was reported to be challenging, as half of the TB patients did not want to participate in a pilot VOT study in northern Viet Nam despite the availability of free smartphones and technical support [21]. Considering the low rates of TB treatment completion in the private sector, many organizations through the WHO have partnered with National TB Programs to engage with the private sector and encourage the adoption of PPM-DOTS by offering free anti-TB drugs and support personnel [47]. The outcomes of these studies showed that PPM-DOTS was accepted by many private providers and was successful in improving treatment outcomes; however, challenges with using PPM-DOTS included large-scale implementation costs, increased burden on private providers and uncertainties about sustainability [42,48–51]. From a programmatic perspective, increasing performance-based incentives to prioritize treatment adherence and completion through continued PPM projects may raise provider interest in VOT utilization [52,53]. As PPM VOT projects have never been piloted, our study suggests that the widespread use of VOT can act as a facilitator for standardized treatment monitoring, convenient patient follow up, and improved treatment outcomes in the private sector [18].

Our study had several limitations. The policy environment for private sector TB treatment in Viet Nam is highly restricted and providers are expected to refer TB patients to the public sector. Currently, there are no published data on private provider perceptions of VOT for TB treatment in Viet Nam or elsewhere, so we were unable to verify the validity of the mean difference and population variance estimates in our sample size calculation and therefore may have underpowered the study. Practicing in the private sector, especially for TB treatment, is sensitive and many private providers do not want to be identified. Therefore, recruitment of private providers who were actively treating TB patients was a challenge given that many of these providers exposed themselves to further investigation by local authorities through their participation. As such, this may have resulted in selection bias. Specifically, participating providers may have had a greater risk appetite, which could have translated to greater acceptability of novel and unproven technologies. Conversely, the substantial proportion (22%) of the private providers we approached that refused to be surveyed would likely have had less favorable attitudes toward VOT including those who may not be open to new technology, unknown to our network, or would not like others to know about their practice. In addition, we employed chain-sampling which has an inherent social desirability bias based on the individual who introduced the interviewer to the provider. As such, our results likely overestimate the acceptability of VOT and may not be representative of all private practitioners providing TB care in Viet Nam as a whole. Nevertheless, we were able to approach private providers with a large volume of patients, whom we consider the target population for VOT implementation. We believe that

the ability to engage these high-volume private providers enhances the generalizability of these findings at least in the Vietnamese context, as these often also represent key opinion leaders in the local context. Our surveys were not conducted by the same individual in each city; thus, an interviewer bias may have caused varying levels of acceptance across different cities. We attempted to minimize this bias by conducting joint training sessions for study staff of all three cities. Finally, we believe that the inclusion of a qualitative component would have strengthened our study. Although we included a space in several of our questions for write-in explanations, these were seldom utilized. Future studies could benefit from conducting a qualitative study to better understand specific causes of provider attitudes towards VOT.

Despite the aforementioned challenges and limitations, studies examining provider attitudes towards mobile or telehealth technologies are sparse. The findings from our study are relatively specific to private health care in Viet Nam; however, this information is still very relevant to other high-TB burden countries with a large proportion of patients seeking care in the private sector [54]. As such, we believe these results have further elucidated the inherent potential of this technology and may inform its potential to improve quality of care across all stakeholders in TB care and prevention.

## Conclusions

Private providers surveyed across three different cities in Viet Nam expressed positive attitudes towards VOT technology and perceived value in using the VOT within their practice. While our survey captures private provider attitudes about their willingness to use VOT, it does not assess their actual use of the technology. We suggest that further studies examine the feasibility of using VOT in the private sector, as the potential public health impacts of private-sector VOT use could be very beneficial for both programs and TB patients. However, in order for VOT to be feasible, challenges in implementation should be considered such as effectively engaging with private sector doctors and targeting an appropriate patient population.

## Supporting information

**S1 Table. Cronbach's alpha (reliability coefficient).**
(PDF)

**S2 Table. Constructs of acceptability associated with the willingness to use VOT in Ha Noi.**
(PDF)

**S3 Table. Constructs of acceptability associated with the willingness to use VOT in Ho Chi Minh City.**
(PDF)

**S4 Table. Comparison of VOT acceptability among private-sector doctors for Ho Chi Minh City and Ha Noi.**
(PDF)

**S5 Table. Comparison of VOT acceptability among private-sector doctors for Ha Noi and Hai Phong.**
(PDF)

**S6 Table. Adjusting for multiple comparisons using the Holm-Bonferroni sequential correction.**
(PDF)

**S1 Dataset. Private provider VOT survey responses.**
(XLS)

**S1 Survey. Private provider acceptability survey in English.**
(PDF)

**S2 Survey. Private provider acceptability survey in Vietnamese.**
(PDF)

## Acknowledgments

The authors express their sincere gratitude to the Viet Nam National Tuberculosis Control Programme, the Provincial TB Hospitals of the three cities and the staff working at the District TB Units for their support and referral of potential participants. The authors also wish to thank Dr. Giang T. Le, Dr. Thanh N. Vu and the Ho Chi Minh City Public Health Association and all participating private providers. We further acknowledge the contributions of Dr Tran Thi Thuy Dung in interviewing participants and the assistance of Kelly M. Collins from Sur-eAdhere Mobile Technology.

## Author Contributions

**Conceptualization:** Lan Huu Nguyen, Phuong Thi Minh Tran, Thu Anh Dam, Rachel Jeanette Forse, Andrew James Codlin.

**Data curation:** Phuong Thi Minh Tran, Thu Anh Dam, Andrew James Codlin, Huy Ba Huynh, Thuy Thi Thu Dong.

**Formal analysis:** Phuong Thi Minh Tran, Thu Anh Dam, Rachel Jeanette Forse, Andrew James Codlin, Luan Nguyen Quang Vo.

**Funding acquisition:** Luan Nguyen Quang Vo.

**Investigation:** Lan Huu Nguyen, Huy Ba Huynh, Thuy Thi Thu Dong.

**Methodology:** Lan Huu Nguyen, Phuong Thi Minh Tran, Rachel Jeanette Forse, Andrew James Codlin, Luan Nguyen Quang Vo.

**Project administration:** Lan Huu Nguyen, Rachel Jeanette Forse, Andrew James Codlin, Luan Nguyen Quang Vo.

**Supervision:** Lan Huu Nguyen, Rachel Jeanette Forse, Andrew James Codlin, Giang Hoai Nguyen, Vinh Van Truong, Ha Thi Minh Dang, Tuan Dinh Nguyen, Hoa Binh Nguyen, Nhung Viet Nguyen, Amera Khan, Jacob Creswell, Luan Nguyen Quang Vo.

**Validation:** Huy Ba Huynh, Thuy Thi Thu Dong.

**Writing – original draft:** Lan Huu Nguyen, Phuong Thi Minh Tran, Thu Anh Dam, Rachel Jeanette Forse, Andrew James Codlin.

**Writing – review & editing:** Lan Huu Nguyen, Phuong Thi Minh Tran, Thu Anh Dam, Rachel Jeanette Forse, Andrew James Codlin, Huy Ba Huynh, Thuy Thi Thu Dong, Giang Hoai Nguyen, Vinh Van Truong, Ha Thi Minh Dang, Tuan Dinh Nguyen, Hoa Binh Nguyen, Nhung Viet Nguyen, Amera Khan, Jacob Creswell, Luan Nguyen Quang Vo.

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
