## [Decision Letter · Decision Letter 0]

2 Feb 2021

PONE-D-20-38378

Assessing private provider perceptions and the acceptability of video observed treatment technology for tuberculosis treatment adherence in three cities across Viet Nam

PLOS ONE

Dear Dr. Vo,

Thank you for submitting your manuscript to PLOS ONE. After careful consideration, we feel that it has merit but does not fully meet PLOS ONE’s publication criteria as it currently stands. Therefore, we invite you to submit a revised version of the manuscript that addresses the points raised during the review process.

We look forward to receiving your revised manuscript.

Kind regards,

Kevin Schwartzman

Academic Editor

PLOS ONE

Journal Requirements:

2. Please include a copy of the questionnaire, in the original language, as Supporting Information, or include a citation if it has been published previously.

3. Please provide further details on sample size and power calculations.

4. Please clarify how many hospitals are represented in this sample of 101 providers.

5. In the Methods section, please clarify whether intra-cluster correlation was considered during  estimation of the effective sample size - given that different localities were sampled. Further, in the Statistical analysis section, please elaborate how you accounted for clustering by locality in your statistical models. Did you consider multilevel models.

6. In statistical methods, please refer to any post-hoc corrections to correct for multiple comparisons during your statistical analyses. If these were not performed please justify the reasons. Please refer to our statistical reporting guidelines for assistance (https://journals.plos.org/plosone/s/submission-guidelines.#loc-statistical-reporting).

7.We note that you have indicated that data from this study are available upon request. PLOS only allows data to be available upon request if there are legal or ethical restrictions on sharing data publicly. For information on unacceptable data access restrictions, please see http://journals.plos.org/plosone/s/data-availability#loc-unacceptable-data-access-restrictions.

Additional Editor Comments:

I am providing a second review as a reviewer who had initially indicated their ability to review was unable to do so in a timely manner.

This manuscript addresses the important topic of expansion of TB treatment support using video DOT into the private sector in a middle-income country, i.e. Vietnam.  It involves a quantitative survey of physicians who provide TB care in the private sector, in three Vietnamese cities.

Major Comments:

The manuscript is generally well-written and clear, and provides relevant information to readers.   However, there are some limitations that need to be addressed by the authors.  The first of these relates to the study sample.  The sample size is fairly limited for a quantitative survey (79 respondents), and it is not entirely clear to what extent respondents are representative of those providing TB care within the private sector in the three cities.  For example, many respondents were recruited because of association with the NGO Friends for International Tuberculosis Relief.  It may be that physicians who are associated with this group may be more open to innovation and collaboration than other private sector physicians.  In the Results section, it would be appropriate to clarify how many of the participants had pre-existing relationships with this NGO. In the Introduction, there is some conflation of directly observed treatment (DOT) with the WHO DOTS strategy, of which DOT is one component.  In addition, there is emerging a more nuanced understanding of and approach to treatment observation:  studies and meta-analyses have had some conflicting findings with respect to the benefits of DOT, while TB survivors and advocates have emphasized that “one size fits all” DOT may be neither necessary nor appropriate in all contexts.  It would be relevant to incorporate these elements into the Introduction section, and emphasize that video DOT may allow treatment support and supervision in a more individualized and differentiated manner.Although it cannot be remedied at this time, the inclusion of qualitative data would have strengthened the study substantially, through further explanation of private health provider perspectives on why they might/might not be interested in adopting video DOT.  This should be mentioned as a limitation in the discussion section, and as a next step for future investigation.   It is unclear what the stratified analysis involving Ho Chi Minh City (page 13 and Table 3b) adds to the manuscript.  I would suggest removing the text on lines 224-229 and Table 3b.  This could be moved to the appendix if the authors wish.

Minor Comments:

In the tables, please round all percentages to the nearest whole number.  Two decimal places are unnecessary.

Reviewers' comments:

Reviewer's Responses to Questions

**Comments to the Author**

1. Is the manuscript technically sound, and do the data support the conclusions?

Reviewer #1: Yes

2. Has the statistical analysis been performed appropriately and rigorously? 

Reviewer #1: N/A

3. Have the authors made all data underlying the findings in their manuscript fully available?

Reviewer #1: Yes

4. Is the manuscript presented in an intelligible fashion and written in standard English?

Reviewer #1: Yes

5. Review Comments to the Author

Reviewer #1: A brief, but useful/appropriate intro (background on TB, DOTs etc) and reasons for carrying out research. However, though the authors chose to focus on private providers, they don't really go into reasons for this. Is the public sector already using VOT? Would it produce different findings (i.e. would VOT be more acceptable)? Or did the authors just think this is a good way of involving private practice?

Fair description of the research - sampling, chosen methods, data collection and analysis all described and discussed. Their literature search missed a couple of important papers – most likely as this research was carried out before publication?

The instrument they devised for responses appears reasonable and clearly demonstrated some internal validity - It may be worth justifying the development of this tool in the absence of any internationally validated instrument.

Somewhat limited description of what VOT is and the type of VOT proposed in the study (using the sureadhere app, so asynchronous). I was able to follow, but not sure I could have if I didn't know about the sureadhere app.

Gaining the opinion of providers/services on the use of VOT is important, and, as the authors say, there is a dearth of research in this area of. Description of the findings/results, is fairly specific to private healthcare in Vietnam. It would be helpful to comment that the findings, while not directly applicable, are relevant to other high burden settings.

6. PLOS authors have the option to publish the peer review history of their article (what does this mean?). If published, this will include your full peer review and any attached files.

Reviewer #1: No

---

## [Author Response · Author response to Decision Letter 0]

18 Mar 2021

Dear Kevin Schwartzman and Reviewer 2, 

Thank you very much for your thorough review and thoughtful comments. A letter that responds to each point raised has been uploaded in a separate file labeled "Response to Reviewers".

---

## [Editor Report · Decision Letter 1]

22 Mar 2021

PONE-D-20-38378R1

Assessing private provider perceptions and the acceptability of video observed treatment technology for tuberculosis treatment adherence in three cities across Viet Nam

PLOS ONE

Dear Dr. Vo,

Thank you for submitting your manuscript to PLOS ONE. After careful consideration, we feel that it has merit but does not fully meet PLOS ONE’s publication criteria as it currently stands. Therefore, we invite you to submit a revised version of the manuscript that addresses the points raised during the review process.

Thank you for submitting your revised manuscript. I appreciate the many updates you have provided in response to earlier comments, most of which have now been handled appropriately. In my view there are two remaining areas for improvement.

The first of these is your approach to multiple comparisons, which was raised by journal review staff. While I appreciate the use of the residual analysis, I think a more appropriate approach would be to correct directly for multiple comparisons (e.g. Bonferroni). In table 2, the repeated use of "**" to label variables deemed most responsible for differences between cities doesn't add much, as nearly all variables are labeled as such.

A more minor point is that in tables 1 and 2, the rightmost column is labeled "chi-square test." In fact, as before this column displays P-values in both tables, and should be labeled as such i.e. "P-value." [In fact, in table 2, many of the P-values reflect the Fisher exact test rather than the chi-square test].

We look forward to receiving your revised manuscript.

Kind regards,

Kevin Schwartzman

Academic Editor

PLOS ONE
---

## [Author Response · Author response to Decision Letter 1]

5 Apr 2021

Dear Dr. Kevin Schwartzman, 

Thank you very much for your comments. A point-by-point response is provided in the filed named "Response to Reviewers".

---

## [Editor Report · Decision Letter 2]

12 Apr 2021

Assessing private provider perceptions and the acceptability of video observed treatment technology for tuberculosis treatment adherence in three cities across Viet Nam

PONE-D-20-38378R2

Dear Dr. Vo,

We’re pleased to inform you that your manuscript has been judged scientifically suitable for publication and will be formally accepted for publication once it meets all outstanding technical requirements.

Kind regards,

Kevin Schwartzman

Academic Editor

PLOS ONE

Additional Editor Comments (optional):

Thank you for submitting your revised manuscript and your response to the earlier comments, which I believe you have properly addressed.
---

## [Editor Report · Acceptance letter]

27 Apr 2021

PONE-D-20-38378R2 

Assessing private provider perceptions and the acceptability of video observed treatment technology for tuberculosis treatment adherence in three cities across Viet Nam 

Dear Dr. Vo:

I'm pleased to inform you that your manuscript has been deemed suitable for publication in PLOS ONE. Congratulations! Your manuscript is now with our production department. 

Kind regards, 

on behalf of

Dr. Kevin Schwartzman 

Academic Editor

PLOS ONE